# Copper causes reduced nitrogen fixation but does not accumulate in the nodules of the legume *Lotus japonicus*

**Kathryn Lamoureux, Sheila M. Macfie**◯*

Department of Biology, University of Western Ontario, London, Ontario, Canada

* smacfie@uwo.ca

## Abstract

Copper can be a soil contaminant at concentrations that are toxic to plants, particularly because of copper-induced oxidative stress. The legume-rhizobia partnership that allows for biological nitrogen fixation is sensitive to oxidative stress, and this study investigates if copper acts directly on the machinery of nitrogen fixation, or indirectly via toxicity to the entire plant. When *Lotus japonicus* inoculated with its rhizobial partner *Mesorhizobium loti* was exposed to 300 or 450 µM of copper, biomass was reduced by 30–40% in the shoots, 40–55% in the roots, and 40–60% in the nodules relative to control. While concentrations of copper in shoots and roots increased in proportion to the amount of copper in the growth medium, concentrations of copper in nodules did not vary in response to copper treatment. Malondialdehyde, a marker of oxidative damage, in the nodule similarly did not vary with copper treatment. However, nitrogen fixation and ascorbate peroxidase activity decreased by 40–45% and 40–60%, respectively, which can be indicators of early nodule senescence. This would suggest that copper-induced reduction in nodule activity is not directly due to oxidative stress in the nodules; it is due to stress on the host plant that limits its ability to support its symbionts.

## Introduction

Legumes are an important source of usable nitrogen (N) in terrestrial ecosystems because of their symbiotic association with rhizobia that fix atmospheric $N_2$ to ammonia [1]. Legumes provide direct and indirect benefits to agroecosystems when used in crop rotation, intercropping, or as a cover crop [2,3], and play an important role in the recovery of marginal or polluted lands [4–6]. Legumes house symbiotic rhizobia in specialized root structures called nodules [1]. Nodulation occurs in response to plant-rhizobia signaling, after which the rhizobia begin expressing the enzyme nitrogenase and fixing $N_2$ for the plant in exchange for photosynthates. Nodulation ultimately involves two tightly coordinated processes: organogenesis of the nodule and controlled infection by the rhizobia [7]. Rhizobia are deposited from "infection

**Data availability statement:** All relevant data are within the paper and its Supporting Information files.

**Funding:** This work was supported by the Canadian Natural Science and Engineering Research Council Discovery Grants Program (https://www.nserc-crsng.gc.ca/; grant RGPIN-2017-04229 to SMM). The funders had no role in study design, data collection and analysis, decision to publish, or preparation of the manuscript.

**Competing interests:** The authors have declared that no competing interests exist.

threads" into host cells via an endocytosis-like mechanism [8]. The membrane-bound rhizobia differentiate into bacteroids, collectively referred to as the symbiosome, which acts like an $N_2$-fixing organelle in the host cell [9–11]. Nodules primarily fall under two morphologies: indeterminate and determinate [11,12]. Indeterminate nodules are characterized by persistent meristematic activity and create cylindrical nodules with specific zones of activity. Determinate nodules (such as those seen in *L. japonicus*) lose meristematic activity after rhizobial infection and an initial period of cell division, forming spherical nodules with fairly homogenous inner tissue [11,13].

Nodule senescence is a regulated process that can be considered the final step in the lifespan of the nodule [14,15]. $N_2$ fixation activity peaks at approximately 4 weeks, after which activity steadily declines [14,15]. Symbiosome membrane disintegration begins and lytic symbiosome compartments are formed in order to facilitate nutrient remobilization and reabsorption processes, ultimately ending in the death of the host cell [16]. In unstressed nodules, cells senesce sequentially from oldest to youngest, meaning that in indeterminate nodules it progresses from zone IV outwards, and progresses radially from the center in determinate nodules [10,15]. In determinate nodules, bacteroids do not terminally differentiate and are released during senescence, able to revert to a free-living lifestyle [10]. Rapid early senescence can be induced when the production of sugars by the host plant is limited due to stressors such as defoliation, phosphorus deficiency, darkness and drought [reviewed in 15], or when available N in soil increases [14]. Senescence is generally characterized by activation of proteolytic enzymes, a decrease in cytosolic proteins [10], and by shifts to the nodule redox balance due to the formation of reactive oxygen species (ROS) and changing activity of nodule antioxidants [16].

Nitrogen fixation via nitrogenase is energetically expensive for the host plant, consuming the energy equivalent of 12–18 ATP per molecule of inorganic N assimilated into amino acids [17], so new nodule formation must be regulated to maintain equilibrium between energy production via photosynthesis in the shoots and energy consumption in the nodules [13]. This is controlled via a root-shoot-root feed-back signaling loop called autoregulation of nodulation [12,18]. The high energetic demand of $N_2$ fixation also means that bacteroids must respire aerobically to maintain nitrogenase activity; however, the nitrogenase enzyme is extremely sensitive to $O_2$ [19]. Adaptive mechanisms put in place by the host cell include nodule architecture that limits $O_2$ diffusion into central infected regions [19,20]; high rates of respiratory $O_2$ consumption in the bacteroids, production of the $O_2$-binding leghemoglobin (Lb) [20,21], and the abundant expression of antioxidants [22]. These combine to maintain the near-anaerobic environment (<50 nM $O_2$ in infected cells) conducive to nitrogenase function [23].

Leghemoglobin plays a pivotal role in controlling $O_2$ levels in the nodule by facilitating the steady movement of $O_2$ to the symbiosomal membrane [24]. The presence of Lb in the nodules is closely associated with nodule function because of its role shuttling $O_2$ to the bacteroids [25]. The correlation between the absence of Lb, elevated free $O_2$ concentrations, and impaired nitrogenase activity leading to reduced $N_2$

fixation was demonstrated with RNA interference lines of *Lotus japonicus* [24], and more recently with CRISPR/Cas9 *Lb* gene knockouts [26]. Leghemoglobin degrades during nodule senescence [15].

Nodule antioxidants are similar to those found in the rest of the plant but are present at higher concentrations to compensate for $O_2$ produced during the highly energetic process of $N_2$ fixation [27]. The ascorbate-glutathione pathway is one of the chief antioxidant pathways of the nodule [28], and its associated enzyme ascorbate peroxidase (APX) is a powerful substrate for $H_2O_2$ scavenging [29]. Ascorbate peroxidase is particularly abundant in the nodule (0.9% of total soluble protein) and may be part of the nodule's oxygen diffusion barrier [30]. Depending on species, activity of APX in the nodules can be several times higher than in the adjacent host roots [20,31].

The heavy metal copper (Cu) is used extensively in industrial and agricultural activities. Mining and smelting, fuel combustion, industrial waste, and the use of Cu-based pesticides and fungicides [32,33] have resulted in significant diffuse emission of Cu into soils around the world [34]. Soil Cu content near smelting operations was as high as 875 mg/kg at the Copper Range Smelter in Michigan, USA [35], 548.3 mg/kg at the Guixi Copper Smelter in Jiangxi, China [36], and 1330 mg/kg in Sudbury, Canada [37]. Copper is not highly mobile in soil, so it persists after its introduction to an environment and can accumulate to dangerous levels [33]. Cu is also redox active, meaning that Cu ions catalyze the production of damaging free oxygen radicals [38,39].

Oxidative stress is common among plants exposed to excess Cu [40–43], where Cu-induced ROS cause lipid peroxidation and the formation of cytotoxic byproducts [39] that damage DNA, proteins, and cell membranes [38]. The roots are the first site of contact for Cu and therefore the primary target for Cu-induced damage [44]. Excess Cu impairs growth and elongation, deforms the root structure [45, 46] and reduces the development of root hairs [47]. One of the primary aboveground signs of Cu toxicity is chlorosis [48,49]. Copper-induced ROS formation causes direct damage to photosynthetic activity and interferes with the chloroplast and thylakoid membranes, which increases ROS production as electron transport is interrupted [41,50]. The legume-rhizobia interaction is sensitive to metal-induced stress, and several studies show that nodulation and $N_2$ fixation are decreased in Cu-contaminated soils [45,49,51–56]. Alfalfa seedlings exposed to 0–400 mg/kg Cu had reduced biomass, tissue N concentrations, and nodule number and weight at higher Cu concentrations [51]. The antioxidant response to Cu-induced ROS includes increased activity of APX [51,52]. Soybean grown in 0–100 μM Cu had reduced nodule number and size, and a similar dose-dependent response in antioxidants in response to Cu [53]. Some reviews state that nodules are potential accumulators of heavy metals [16,57] but do not provide measurements of Cu concentrations in nodules specifically. Studies concerned with phytoremediation by legumes report that nodulation increases heavy metal uptake in the host plant, but generally only measure Cu accumulation only in the roots and shoots [46,48,58]. These studies also do not explore a mechanism for the observed decrease in $N_2$ fixation in response to Cu. How Cu toxicity manifests in a wide range of legume species, and the mechanisms by which it inhibits $N_2$ fixation and nodulation are important parts of understanding the impact of Cu pollution on the surrounding ecosystem. The aim of this study is to determine if the detrimental effect of Cu on nodulation and $N_2$ fixation is due to direct oxidative damage to the nodule interior, or if it is due to reduced health of the whole plant and its rhizobia.

## Results

### Copper exposure reduces tissue biomass

Copper exposure reduced both nodule formation and the function of nodules that successfully formed, as well as the biomass of both shoots and roots in *Lotus japonicus*. The experimental Cu concentrations, 300 μM and 450 μM Cu, resulted in a 30% and 40% reduction, respectively, in fresh shoot biomass compared to controls (Fig 1A), and a 10% and 18% reduction in fresh shoot length (Fig 1B). The Cu treatments also caused noticeable interveinal chlorosis in the leaves and browning of leaf edges (S2 Fig A-C). Fresh root mass was reduced by 40% at 300 μM Cu and 55% at 450 μM Cu (Fig 1C) compared to controls, and fresh root length decreased by 27% at 300 μM Cu and 45% at 450 μM Cu (Fig 1D). The plants in the Cu treatment groups also appeared to have reduced lateral root growth and branching compared to control plants, and brown discolouration on the

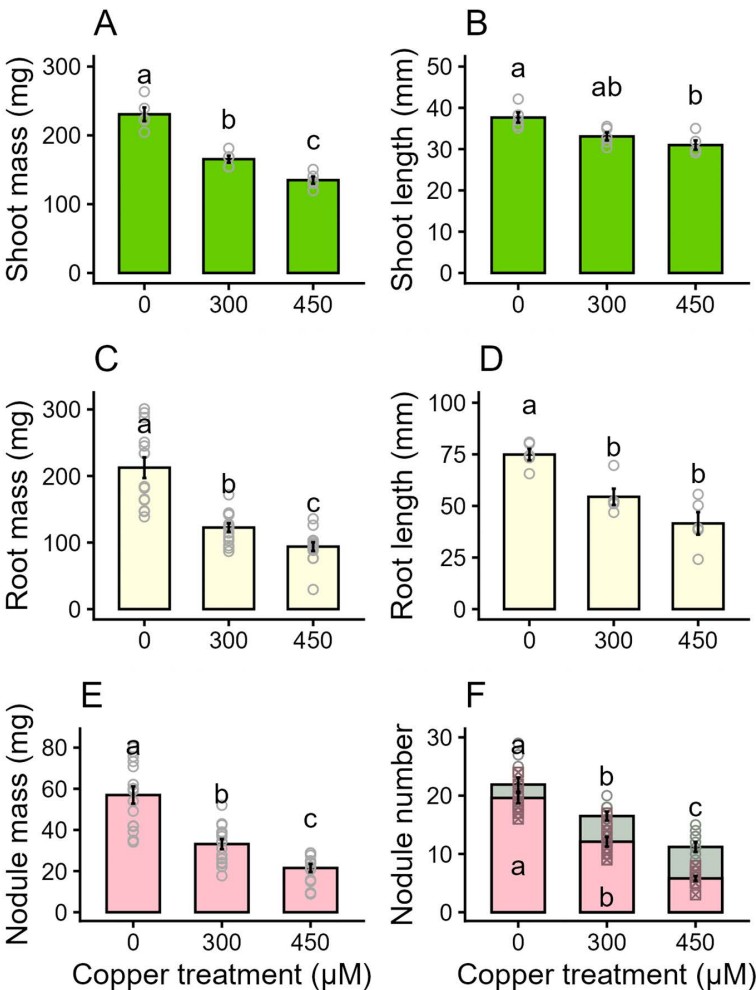

**Fig 1. Fresh tissue biomass and number of nodules per plant.** The average fresh mass (mg) of shoots, roots, and nodules, the length (mm) of roots and shoots, and the number of nodules on *Lotus japonicus* in the control (0 µM copper), 300 µM copper, and 450 µM copper treatments added to basal nutrient solution. Each replicate represents the average of multiple plants in a single pot. Plants were harvested at 42 days post germination and 28 days post inoculation. (A) Fresh shoot mass (n = 5). (B) Fresh shoot length (n = 5). (C) Fresh root mass (n = 14). (D) Fresh root length (n = 5). (E) Fresh nodule mass (0 and 300 µM Cu n = 15, 450 µM Cu n = 13). (F) Mean nodule count, showing the number of active (pink) compared to inactive (green-grey) nodules out of that total count (n = 10). Error bars indicate SEM. Letters indicate significant differences between groups (p < 0.05). Nodule mass was transformed prior to analysis to meet test assumptions. Shoot mass, nodule mass, shoot and root lengths, and nodule counts were analyzed via one-way ANOVA followed by a Tukey's HSD post-hoc test. Root mass was analyzed via Welch's unequal variances test followed by a Games-Howell post-hoc test.

surface of the root (S2 Fig D-F). Total fresh nodule mass was the most sensitive to Cu stress and dropped by 40% at 300 µM Cu and 60% at 450 µM Cu (Fig 1E). The number of nodules also decreased by 25% at 300 µM Cu and 50% at 450 µM Cu, indicating a reduction in successful nodule formation. The Cu treatment groups generally had smaller and fewer nodules than the control group (S2 Fig G-H) and had a higher percentage of inactive green nodules to active pink nodules (Fig 1F, S2 Fig I).

## Copper reduces nitrogen fixation and leghemoglobin content of nodules

An acetylene reduction assay was used to measure the activity of the nitrogenase enzyme in response to Cu. The reduction of acetylene to ethylene decreased in response to increased Cu. When expressed as the nmol of ethylene produced

per plant (Fig 2A), plants in the 450 µM Cu treatment group had 44% decrease in nitrogenase activity compared to control plants, and the 300 µM Cu treatment group had 39% decrease in nitrogenase activity. When the data were adjusted for root mass per pot (Fig 2B) the 450 µM Cu group had a 35% reduction in nitrogenase activity compared to control. The 300 µM Cu group had a non-significant (p = 0.0796) 31.5% reduction in enzyme activity. When the data were adjusted to the estimated nodule mass for each Cu treatment group (Fig 2C), there was a 36% reduction in activity in the 450 µM Cu group, and a non-significant (p = 0.0740) 32% reduction in the 300 µM Cu group. Comparing nitrogenase activity of the whole plant to the activity occurring per gram of nodule tissue indicates that the observed reduction in enzyme activity can partially be explained by the reduction in nodule biomass at higher Cu concentrations.

The results of the pyridine hemochromogen assay for nodule Lb concentration are shown in Fig 2D. Compared to controls, nodule Lb content was reduced by 40% in the 300 µM Cu group and 60% in the 450 µM Cu group, mirroring the reduction in nitrogenase activity, and further indicating a loss of nodule function in response to Cu. The colour of individual nodule interiors (depending on the presence or absence of Lb) was also recorded; the reduction in Lb did not appear uniform across the Cu treatment groups or within individual plants of those groups. In the control group the nodules were overwhelmingly pink and all similar in colour; however, the tissue pools for the 300 µM and 450 µM Cu treatment groups were made up of many green nodules, some that appeared to have intermediate amounts of Lb, and a small number of fully pink nodules.

## Copper does not accumulate in the nodule tissues

Inductively-coupled plasma mass spectrometry was used to analyse the Cu concentrations in shoot, root and nodule tissues (Fig 3). Copper concentrations varied among both tissue types and treatment groups. These findings confirm that the Cu was bioavailable in this experimental system, and that increased exposure resulted in higher Cu accumulation in

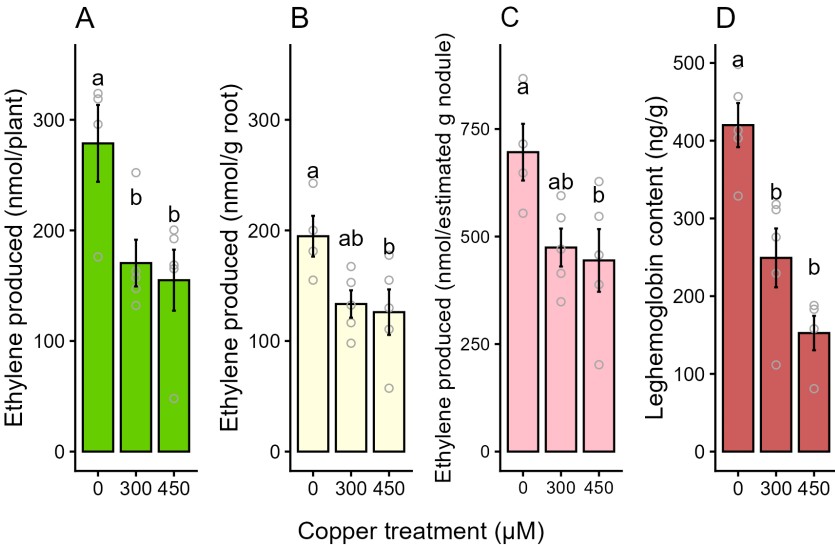

**Fig 2. Acetylene reduction by nitrogenase and nodule leghemoglobin content.** (A-C) Acetylene reduction assay results, showing the nanomoles of acetylene reduced to ethylene over 30 minutes as a proxy for nitrogenase activity in control (0 µM copper), 300 µM copper and 450 µM copper treatments added to basal nutrient solution. (A) Amount of ethylene produced per whole root mass. (B) Ethylene produced per g of fresh root tissue. (C) Ethylene produced per estimated g of nodule tissue. 0 µM Cu group n = 4, other groups n = 5. (D) Results of the colorimetric pyridine hemochromogen assay showing the concentrations of oxygen-binding leghemoglobin (ng/g of fresh nodule tissue) for control, 300 µM copper, and 450 µM copper treatment groups. 450 µM Cu n = 4, all other groups n = 5. Each replicate represents the average of multiple plants from a single pot. Plants were harvested at 42 days post germination and 28 days post inoculation. Error bars indicate SEM, letters indicate significant differences (p < 0.05), analyzed via a two-way ANOVA followed by a Tukey's HSD.

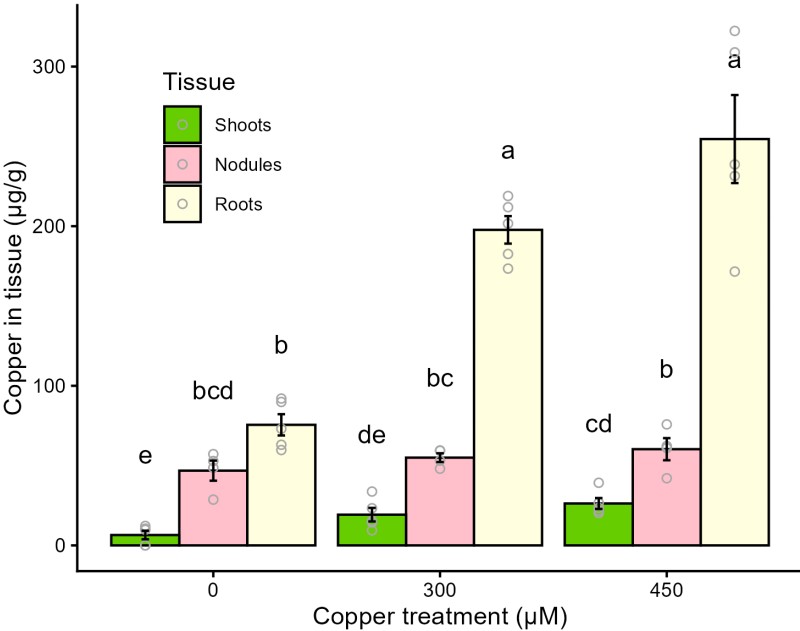

**Fig 3. Copper concentrations in shoot, root and nodule tissues.** The concentration of copper (µg/g of dried tissue) taken up into the shoots (green), nodules (pink) and roots (yellow). Each replicate represents the average of multiple plants in a single pot. Plants were harvested at 42 days post germination and 28 days post inoculation. Treatments added to basal nutrient solution are control (0 µM copper), 300 µM copper, and 450 µM copper. Letters indicate significant differences between groups (p < 0.05). Error bars indicate SEM, n = 5 for roots and shoots, n = 4 for nodules. Data were transformed prior to analysis to meet test assumptions and then analyzed with a two-way ANOVA followed by Tukey's HSD.

plant tissues. Root Cu concentrations increased in both treatment groups, with a 2.5-fold increase relative to control for plants given 300 µM Cu and a 3.4-fold increase for plants given 450 µM Cu. Cu concentrations in the shoots differed only between the control and 450 µM Cu treatment group, in which there was a 4-fold increase relative to control. Root Cu concentrations were the highest and shoot concentrations the lowest regardless of Cu treatment group, with Cu concentrations being 9–11-fold higher in roots than shoots. The nodules had 38–76% less Cu than the roots of the respective treatment groups, and Cu content of the nodules did not differ among treatments, which explains a significant treatment × tissue interaction (p < 0.001).

**Copper causes oxidative damage in root tissues but not nodule tissues**

A thiobarbituric reactive substances (TBARS) assay was used to measure the concentration of the lipid peroxidation end-product malondialdehyde (MDA) in the roots and nodules of the plant as a marker of oxidative damage in the samples (Fig 4A). The concentration of MDA in roots increased by 58% in the 450 µM Cu treatment group relative to the control group, indicating an increase in oxidative damage to those tissues. Roots of the 300 µM Cu group had a non-significant (p = 0.14) 39% increase in MDA concentration. The tissue × treatment interaction was not significant (p > 0.05). Although MDA was also detected in the nodules, the concentration did not vary with Cu treatment. This, combined with the finding that nodule Cu content did not change with Cu treatments (Fig 3), suggests that Cu does not directly affect activity within the nodule via oxidative damage. It is possible that any Cu-induced oxidative stress is mitigated by increased activity of the nodule's antioxidant systems prior to observable membrane damage. To determine if this was the case, the activity of the ROS-scavenging enzyme APX was measured in the nodules, as well as in the roots for comparison to a tissue where it has been established that oxidative damage is occurring (Fig 4B). Activity of APX did not vary significantly

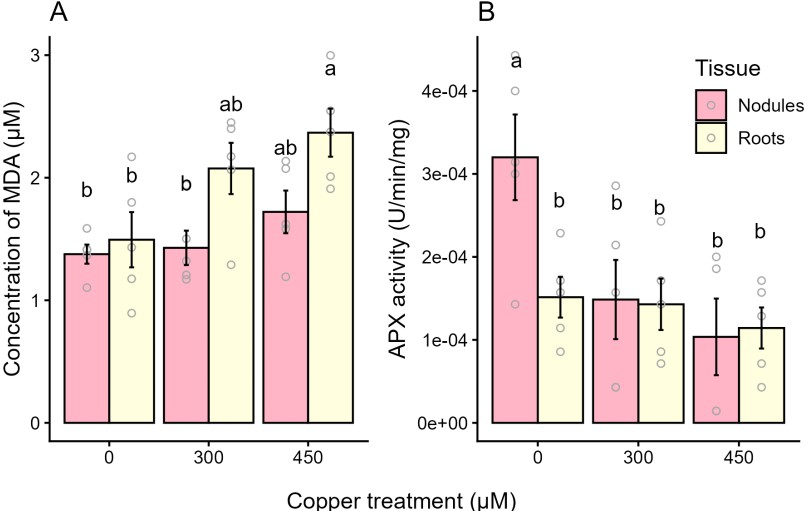

**Fig 4. Concentration of malondialdehyde and activity of ROS-scavenging enzyme ascorbate peroxidase in root and nodule tissues.** (A) Thiobarbituric reactive substances assay results showing the μM concentration of the polyunsaturated fatty acid decomposition product malondialdehyde (MDA) in nodules (pink) and roots (yellow) as an estimation of lipid peroxidation occurring in those tissues. (B) Ascorbate peroxidase (APX) activity in U/min/mg in nodule (pink) and root (yellow) tissues. Treatments added to basal nutrient solution are control (0 μM copper), 300 μM copper, and 450 μM copper. APX 450 μM copper nodules n = 4, all other groups n = 5, Each replicate represents the average of multiple plants in a single pot. Plants were harvested at 42 days post germination and 28 days post inoculation. Error bars indicate SEM, letters indicate significant differences (p < 0.05), analyzed via a two-way ANOVA followed by a Tukey's HSD.

among the root treatment groups, despite signs of Cu-induced stress in those tissues. Activity of APX in the nodules of the Cu-treated plants was 53–67% lower than APX activity in the control group.

## Discussion

The growth of *L. japonicus* and its ability to form nodules was impaired by Cu exposure, as in other legumes [45,49,52]. Nodules were less numerous, smaller, and less active in Cu-treated plants, meaning that Cu interrupted both the plant-rhizobia interactions that initiate the process of nodulation as well as the development and persistence of nodules that did form. Lower nodulation could be the result of the combined effect of Cu on i) root morphology that lowers the number of potential rhizobial infection sites [59], ii) shoot morphology that limits photosynthetic output and increases autoregulation of nodulation signals [60], and/or iii) lower survival of the free-living *M. loti* under Cu stress [61] (S4 Fig).

The tips of emerging root hairs are the primary sites of entry for rhizobia and are important for the formation of infection threads [62,63]. Reduced nodulation was observed in *L. japonicus* mutants that produced no root hairs (*rhl1* and *slp*) [64], mutants with altered biosynthesis pathways critical for root hair development (*dahps1–1* and *dahps1–2*) [65], and plants with thicker and shorter root systems (*rdol*) [66]. Root biomass reduction, particularly fine roots and root hairs, is a characteristic response to excess Cu in many plant species [50,67]. The Cu-induced reduction in root mass seen in our experiment (S3 Fig) contributed to the drop in nodulation, an observation shared by other researchers [56]. Copper-induced inhibition of water and mineral absorption in the roots [68,69] ultimately impairs photosynthesis and limits the production of carbon compounds required to maintain the energetically expensive nitrogenase enzyme. Shoot activity controls nodule formation via the autoregulation of nodulation signalling pathways that maintain homeostasis between carbon production in the leaves and carbon consumption in the nodule [60]. Copper may then indirectly limit new nodule formation due to its ability to damage photosynthetic machinery [48,70]. Because of this reciprocal sink-to-source connection, impaired photosynthetic output can also trigger senescence of established nodules [15].

Total N$_2$ fixation in our experiment was reduced due to the Cu-induced decrease in both the number and biomass of nodules formed, and may also have been directly affected by a drop in nitrogenase activity within the nodules themselves. As nitrogenase occurs exclusively in the bacteroids within the plant's nodules, lower nodule mass naturally results in a reduction in the total nitrogenase activity of the plant. This pattern was also observed in alfalfa under Cu stress, where nodule number was the most sensitive parameter in response to Cu and total nitrogenase activity decreased in response to nodule biomass [71]. Leghemoglobin content is closely associated with N$_2$ fixation [24], and reduced nitrogenase activity and Lb content are both key characteristics of nodule senescence [14,15,72].

Despite the evidence that excess Cu inhibited nodule development and activity, measurements of tissue Cu concentrations revealed that Cu did not accumulate in the nodule tissues, even though the nodules are immediately adjacent to the roots and have a connected vasculature [73]. This would indicate that Cu transport to the nodule tissues is restricted in some fashion so that internal Cu concentration remains relatively constant. Copper triggers an increase in cell wall thickening [74] and the deposition of pectin [75] as well as various phenolic compounds in the roots [72,76,77]. This is thought to reduce the mobility and toxicity of Cu by increasing the amount of Cu that can be sequestered via binding within the cell walls [78]. It is possible that similar protective changes to cell wall composition occurs in the nodule to reduce Cu entry to those tissues. In our study the lack of variability in nodule Cu across the treatments means it is unlikely that the observed decreases in N$_2$ fixation and nodulation are due to direct effects of Cu on the nodule itself, and suggests that Cu toxicity acts indirectly on the nodule via its effect on the rest of the plant. Premature, stress-induced senescence of the nodules due to the detrimental effects of Cu on the roots and shoots could explain this indirect effect.

In biological systems the primary method of toxicity from Cu is oxidative damage [43], and lipid peroxidation in the roots of legumes has previously been reported to increase in response to Cu stress [40,46,51,52,76]. In our study the nodule tissues showed no sign of direct oxidative damage despite toxicity appearing in the immediately adjacent roots, which further suggests that Cu is not acting directly on the nodule tissues. Lipid peroxidation in nodules is often reported as a symptom of both developmental and stress-induced senescence [16], and concentrations of MDA generally appear to increase in the nodules during developmental senescence [79,80]. Measurements of MDA during stress-induced senescence are mixed; dark stress [81] and salt stress [79] induced senescence but did not cause lipid peroxidation, while exogenous applications of Cd [82] and H$_2$O$_2$ [79] did cause an increase in MDA. Membrane damage, particularly to the symbiosome, generally occurs early in the process of senescence [14]. Loscos et al. [79] measured lipid peroxidation and the accumulation of oxidized proteins in *Phaseolus vulgaris* nodules that were actively fixing N$_2$, mid-senescence, and fully senesced. Concentrations of MDA in the group that was actively undergoing senescence increased by 69% compared to the non-senescent control group, whereas MDA in the fully senescent group was the same as control. However, accumulation of oxidized protein was 3.5-fold higher in the fully senescent nodules compared to control, indicating that oxidative changes had at one point occurred. This, along with the varying results in lipid peroxidation levels from other studies, implies that timing is an important facet of measuring oxidative stress in nodules. In our study it is likely that lipid peroxidation did occur shortly after Cu exposure while the nodules of the treatment groups were still semi-active, but evidence of this was not captured.

The difference in APX activity between the control and Cu-treated nodules may also be explained by premature loss of enzymatic activity during stress-induced senescence. The transcript levels of APX are expected to be at least 10-fold higher in the nodules than in adjacent infected roots [20,31] in order to protect the nodule from ROS produced during bacteroid respiration [83]. This would indicate that the high APX activity observed under control conditions is typical for the nodule, and loss of activity may be an indicator of plant stress. Loss of APX activity during developmental senescence was observed in the nodules of *P. vulgaris*; however, no differences in APX activity in response to either Cd or exogenous H$_2$O$_2$ application were reported [79]. No studies could be found that measured APX or any other antioxidant in the nodule in response to Cu specifically, and the narrow scope of antioxidant responses tested in this study mean that definitive conclusions about the nodule redox state cannot be drawn without further research. The roots in the Cu treatment groups

in our study were likewise expected to show increased ROS scavenging activity due to the increase in both Cu concentration and lipid peroxidation levels in those tissues, but showed no significant changes across all treatment groups. Similar results were observed in response to Cu in the roots of some other legumes; in one study on alfalfa, root APX activity decreased with increasing Cu [46] and in another it increased at low (0–50 µM) concentrations before dropping again at around 100 µM Cu [52]. No significant difference in root APX activity across Cu treatments was observed in *M. lupulina* [54]. Critically, in the above studies, the activities of other antioxidant enzymes such as SOD and CAT increased in response to Cu stress, so it is likely that our study did not capture a wide enough range of antioxidant activity for a fulsome picture of antioxidant responses to Cu-induced ROS in the roots and nodules.

Our study provides novel information about Cu stress in the nodules of legumes by determining that Cu-induced stress acts indirectly on nodules to reduce $N_2$ fixation. Differences in nodulation and $N_2$ fixation observed were not due to accumulation of Cu in the nodule interior and Cu did not appear to cause oxidative damage directly to the nodule tissues. Copper-induced stress in the roots and shoots (e.g., reduced root surface area for water and nutrient uptake and chlorosis of the leaves) negatively affected the plant's ability to both form nodules and to successfully photosynthesize. This likely had a severe impact on the plant's ability to energetically support many rhizobia and triggered premature nodule senescence.

## Materials and methods

### Plant and rhizobial materials, growth conditions

*Lotus japonicus* ('Gifu' accession number B-129) plants were grown from seed and inoculated with *Mesorhizobium loti* (rhizobial strain NZP2235). Both seeds and rhizobia were provided by the London Research and Development Centre – Agriculture and Agri-Food Canada. All plant cultivation occurred in a controlled environment chamber (custom built chamber, CAN-TROL Environmental Systems Limited) set at 22°C, 65–70% relative humidity, 16/8 day/night, 197 ± 7 µmol/sec/m. *Lotus japonicus* seeds were scarified with sandpaper and surface-sterilized with 70% EtOH/0.1% sodium dodecyl sulfate, followed by 1.2% sodium hypochlorite/0.1% sodium dodecyl sulfate, then rinsed 10 times in sterile reverse osmosis (RO) water. Seeds were left to germinate on filter paper (Whatman 230 90 mm, Cytiva) soaked with sterile RO water in sealed Petri dishes for 7 days. Seedlings of uniform size were then selected and transferred to a sand-hydroponic culture system, consisting of 12.7 cm diameter pots (8 seedlings/pot) filled with sterile course-grain sand (1.5–2 mm) that was saturated with a modified low-nitrogen Broughton and Dilworth nutrient solution [84] (S1 Table). At 14 days post-germination each pot was saturated with 35 mL of nutrient solution spiked with 0, 300 or 450 µM additional $CuSO_4$. Concentrations of Cu were chosen to induce sub-lethal toxicity based on dose-response experiments (S1 Fig). These values are well below the limit for Cu content in agricultural soil in Canada, 63 mg/kg [85]; in our sand-culture system, 450 µM is equivalent to 5.7 mg Cu per kg sand.

Shortly after Cu application seedlings were each inoculated with 1 mL of *M. loti* suspension pipetted directly to the base of each plant (8 mL total per pot). Prior to the inoculation day, *M. loti* was cultured in sterile liquid tryptone-yeast medium (pH 6.8) and incubated at 26°C for 72 hours. On plant inoculation day the bacterial culture was centrifuged (model 5804 R, Eppendorf) at 2500 × g for 20 minutes, the pellet was re-suspended in sterile RO water and diluted to $2.06 \times 10^8$ cells/mL. This suspension was applied to the plants. At 42 days post-germination the mature plants were removed whole from the sand-hydroponic system, and root, shoot and nodule tissues were separated. Individual tissues were weighed, the root length was recorded, nodules were counted under a dissecting microscope, and the nodule interior colour was noted. Because the final number of plants in each pot varied from 7 to 8, the average tissue mass, length, and nodule count of the plants in each pot was calculated. The average from each pot was then considered an independent biological replicate.

### Nitrogenase activity

To estimate nitrogenase activity, an acetylene reduction assay was performed at the London Research and Development Centre – Agriculture and Agri-Food Canada, using a combination of methods described in Hayashi et al. [86] and Montes-Luz et al. [87]. Chromatographic separation was achieved by using a GS-GasPro column (30 m × 320 µm × 0.32 mm,

Agilent Technologies). The inlet was held at 260°C and the analysis was conducted isothermally at 90°C. The gas chromatograph flame ionization detector (GC-FID) was operated in split mode at a ratio of 20:1. The FID was set at 320°C, with hydrogen flow at 35 mL/min, air flow 400 mL/min and the makeup flow of 22 mL/min. The flow was set to 1.75 mL/min with a run time of 4.5 min. The plants were de-potted and the 8 root systems were removed, weighed, and placed into a 20 mL scintillation vial sealed with cap and septa. 200 µL of air was pulled from each vial and replaced with 200 µL of acetylene to establish 1% (v:v) acetylene at 1 atm in the vial. Immediately 50 µL of air was then withdrawn from the vial and sent through the GC-FID to measure any ethylene produced by the roots due to stress from manipulation. This first analysis was considered time 0. The vials were then incubated at room temperature for 30 minutes, after which another 50 µL was withdrawn and ethylene content was measured again. The difference in ethylene content between 0 and 30 minutes was considered the amount of acetylene reduced to ethylene by the root systems. A blank vial was also processed in order to account for potential ethylene contamination in the acetylene source, and to determine any minute changes in acetylene content withdrawn from the reservoir tank that would affect mapping the sample data to a standard curve of ethylene concentration to peak-area. Nitrogenase activity per vial (plant) was normalized to both root and nodule mass. While total root mass in each vial was measured, the nodule mass was not. Therefore, nodule mass was estimated using known total root and nodule masses from 15 other plants from each treatment group.

## Leghemoglobin concentration

Leghemoglobin concentrations were determined using a spectrophotometer (Biochrom Ultrospec 2100 *pro*, Harvard Bioscience) following methods originally described in Appleby & Bergersen [88] and the extinction coefficients for hemoglobin *b* determined by Berry & Trumpower [89]. Harvested nodules were flash frozen in liquid nitrogen and ground in a chilled mortar. Depending on available tissue, 0.05 to 0.15 g of the ground tissue was added to a microcentrifuge tube with 700–1500 µL of 0.1 M potassium phosphate buffer (PPB) (pH 7.4). The liquid was filtered into a new tube through 2 layers of 5 cm × 5 cm cheese cloth, then centrifuged for 20 minutes at 10,000 × g and 650 µL of the supernatant was pipetted into a microcentrifuge tube. An equal volume of 0.2 M NaOH/4.2 M pyridine solution was added to the tubes and mixed gently. The resulting hemochrome was divided into cuvettes labelled 'reduced' or 'oxidized'. To the 'reduced' group, approximately 2 mg of sodium dithionite was added and mixed gently to avoid aeration. After 2 minutes the absorbance was read at 539 and 556 nm sequentially, with the spectrophotometer blanked at each wavelength with a 1:1 ratio of 0.1 M PPB and NaOH/pyridine solution and 2 mg of sodium dithionite. To the 'oxidized' cuvettes, approximately 2 mg of potassium hexacyanoferrate was added and absorbance was read at 539 and 556 nm, with the spectrophotometer again blanked at each wavelength with a 1:1 ratio of 0.1 M PPB and NaOH/pyridine solution and 2 mg of potassium hexacyanoferrate. The difference in absorption between reduced and oxidized samples was used to calculate Lb concentration (in mM) using this equation based on Beer's Law:

$$Lb = \frac{\left[(Abs_{556}red - Abs_{556}oxi) - (Abs_{539}red - Abs_{539}oxi)\right]}{\varepsilon \times l} \qquad (1)$$

Where $\varepsilon =$ is the molar extinction coefficient of heme b (23.98/mM/cm) [88] and $\ell$ is the pathlength of the cuvette (1 cm).

## Tissue copper concentration

The United States Environmental Protection Agency test method 3010A [90] as modified by Akhter et al. [91] was used to confirm Cu uptake and determine Cu concentrations in the shoots, roots, and nodules. Harvested tissues were dried at 60°C to a constant weight and crushed into fine pieces (<1 mm). A representative 0.1 g (roots and shoots) or 0.05 g (nodules) sample for each treatment group was digested overnight in metal-free tubes with 1 or 0.5 mL, respectively, of OmniTrace® nitric acid. Regent blanks and a certified reference material (NIST #1750a) were also digested. The samples were

then boiled until fumes ran clear and filtered to remove solids. Samples were diluted with RO water (2% acid by volume) and then sent to A&L Canada Laboratories Inc. (London, Ontario) for Cu content analysis via inductively-coupled plasma mass spectrometry.

### Lipid peroxidation

Tissue was flash-frozen in liquid nitrogen and ground in a chilled mortar immediately after harvesting, then the Cayman Chemicals TBARS assay kit (Item No. 700870) was used to assess lipid peroxidation per the manufacturer's instructions. Duplicate subsamples were pipetted into a 96-well plate and absorbance was read with a spectrophotometer (SpectraMax M5 96-well plate reader, Molecular Devices) set at 535 nm. Absorbance values were compared to a standard curve of MDA in order to quantify amount of MDA in each sample.

### Ascorbate peroxidase activity

The activity of the antioxidant APX in the nodules was measured via spectrophotometric methods (Biochrom Ultrospec 2100 *pro*, Harvard Bioscience) described in Senthilkumar et al. [92]. Roots and nodules were harvested into 50 mM PPB (pH 7.0) over wet ice and ground in chilled mortars. Depending on available mass, between 0.05 and 0.15 g of ground tissue was mixed thoroughly with 500–1500 µL of 50 mM PPB and put into a labelled microcentrifuge tube. Samples were centrifuged for 15 minutes at 11,180 × g at 4°C. Without disturbing the pellet, 0.1 mL of supernatant was pipetted into a cuvette over wet ice, in duplicate. To each cuvette 0.9 mL of reaction mixture containing 50 mM PPB, 0.5 mM ascorbic acid, 1.0 mM $H_2O_2$ and 0.1 M ethylenediaminetetraacetic acid was added, then the decrease in absorbance at 290 nm was read every 15 seconds for each sample for 90 seconds. The spectrophotometer was blanked with a 1:9 ratio of 50 mM PPB to reaction mixture without $H_2O_2$. The difference in absorbance at 90 seconds and 15 seconds was used to calculate the unit activity in U/min/mg using this equation:

$$\textit{Unit activity} = \left( \Delta \text{Abs}_{290} \right) \left( \frac{V_r}{\varepsilon} \right) \left( V_t \right)$$

(2)

where $\Delta \text{Abs}_{290}$ is the change in absorbance over 75 seconds, $V_r$ is the total volume of reaction mixture (1 mL), $V_t$ is the volume of sample taken for analysis (0.1 mL), and $\varepsilon$ is the extinction coefficient of ascorbate, 2.8/mM/cm. Reported activity was adjusted to per minute.

### Statistical analysis

All statistical analyses were done with R (version 4.3.1) via RStudio (Version 2023.06.1 + 524) using the tidyverse, rstatix, car and ggpubr packages. All graphs were produced in R using the ggplot2, cowplot and multcompView packages. A one-way ANOVA was used to assess the effect of Cu treatment on the response variables in experiments that compared a single tissue type, and a two-way ANOVA was used for experiments that compared between tissue types. When significant main effects were detected, ANOVA was followed by a Tukeys Test. Normality and homogeneity of variance were verified in the raw data for all but three variables: tissue Cu concentration data were transformed by taking the square root prior to analysis, nodule biomass data were transformed by the arcsine square root function prior to analysis, root biomass data were analyzed with a Welch's unequal variance ANOVA followed by a Games-Howell post-hoc test.

### Supporting information

**S1 Table. Nutrient solution recipe modified from Broughton and Dilworth [84].**
(PDF)

**S1 Fig. Dose-response of *Lotus japonicus* to 0–1250 μM copper (in addition to the basal concentration of copper in B&D nutrient solution).** (A) Shoot length. (B) Shoot mass (mg). (C) Root length. (D) Root mass. (E) Number of nodules per plant. (F) Nodule mass. Error bars indicate SEM, n = 4 for all tissues. Letters indicate significant differences between groups ($p < 0.05$), analyzed with a one-way ANOVA followed by a Tukey's HSD, with the exception of root mass data, which were analyzed with a Kruskal-Wallis test followed by Dunn's pairwise comparisons because the data failed the homogeneity of variance test.
(PDF)

**S2 Fig. Photographs of representative shoot, root and nodule samples.** (A – C) Whole shoots, with a single trifoliate leaf laid flat to show leaf colour. (D – F) Root systems with nodules removed. Left to right are the control (0 μM copper), 300 μM copper, and 450 μM copper treatments added to basal nutrient solution. G) Nodules attached to roots of the control group. H) Nodules attached to roots of the 450 μM copper group. I) Representative nodules of the control (right) and 450 μM copper (left) treatment groups cut open to show interior colour differences. All bars are 5 mm.
(PDF)

**S3 Fig. Nodule number plotted against fresh root mass.** Fresh root mass (g) plotted against nodule number in 0 μM copper (red), 300 μM copper (green) and 450 μM copper (blue) treatments added to basal nutrient solution.
(PDF)

**S4 Fig. Dose-response of *Mesorhizobium loti* to 0–500 μM copper.** Cells/mL of *M. loti* at 0–500 μM copper (in addition to the basal concentration of copper in B&D nutrient solution). Data were collected over 5 days (day 0 data not included). Error bars indicate SEM, n = 3.
(PDF)

**S1 Data. Data used in this article.**
(XLSX)

## Acknowledgments

We are grateful to Krzysztof Szczyglowski and Tim McDowell at the London Research and Development Centre – Agriculture and Agri-Food Canada for supplying *L. japonicus* seeds and rhizobial culture, and assisting with the acetylene reduction assay and associated analyses.

## Author contributions

**Conceptualization:** Kathryn Lamoureux, Sheila M. Macfie.

**Data curation:** Kathryn Lamoureux.

**Formal analysis:** Kathryn Lamoureux, Sheila M. Macfie.

**Funding acquisition:** Sheila M. Macfie.

**Investigation:** Kathryn Lamoureux.

**Methodology:** Sheila M. Macfie.

**Project administration:** Sheila M. Macfie.

**Resources:** Sheila M. Macfie.

**Supervision:** Sheila M. Macfie.

**Validation:** Kathryn Lamoureux.

**Visualization:** Kathryn Lamoureux.

**Writing – original draft:** Kathryn Lamoureux.

**Writing – review & editing:** Kathryn Lamoureux, Sheila M. Macfie.

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
