## [Decision Letter · Decision Letter 0]

30 Sep 2025

PONE-D-25-29926Copper-induced stress in the legume Lotus japonicus causes loss of nitrogen fixation but not oxidative damage in nodulesPLOS ONE

Dear Dr. Macfie,

Thank you for submitting your manuscript to PLOS ONE. After careful consideration, we feel that it has merit but does not fully meet PLOS ONE’s publication criteria as it currently stands. Therefore, we invite you to submit a revised version of the manuscript that addresses the points raised during the review process.

We look forward to receiving your revised manuscript.

Kind regards,

Ying Ma, Ph.D.

Academic Editor

PLOS ONE

Journal Requirements:

2. We note that there is identifying data in the Supporting Information file <S1 Table.docx>. Due to the inclusion of these potentially identifying data, we have removed this file from your file inventory. Prior to sharing human research participant data, authors should consult with an ethics committee to ensure data are shared in accordance with participant consent and all applicable local laws.

-Location data

Please remove or anonymize all personal information (Name), ensure that the data shared are in accordance with participant consent, and re-upload a fully anonymized data set. Please note that spreadsheet columns with personal information must be removed and not hidden as all hidden columns will appear in the published file.

Reviewers' comments:

Reviewer's Responses to Questions

**Comments to the Author**

1. Is the manuscript technically sound, and do the data support the conclusions?

Reviewer #1: Partly

Reviewer #2: Partly

2. Has the statistical analysis been performed appropriately and rigorously? 

Reviewer #1: Yes

Reviewer #2: I Don't Know

3. Have the authors made all data underlying the findings in their manuscript fully available?

Reviewer #1: Yes

Reviewer #2: Yes

4. Is the manuscript presented in an intelligible fashion and written in standard English?

Reviewer #1: Yes

Reviewer #2: Yes

5. Review Comments to the Author

Reviewer #1: The manuscript submitted for review is carefully written and engaging. The research topic is current and addresses an important issue related to the increasing environmental pollution with metals such as copper and the legume-rhizobia symbiosis, which is important for ecology and agriculture.

However, the manuscript requires corrections. Below are a few comments:

Introduction - the section is well-written and explains the aspects studied.

Line 32 - the ability to fix nitrogen should be clarified as being characteristic of bacteroids, not plants.

Results - the results regarding the determination of MDA and APX for shoots are missing, as these results would provide a broader, more accurate answer to the statement in the title of the paper.

Methods:

Please expand abbreviations used for the first time, especially for reagents, buffers, etc., e.g., PPB. Please add the equipment used for analyses.

Graphs - are prepared in an appropriately legible manner.

Reviewer #2: In this study, Lamoureux et al. investigated the molecular effects of copper treatment on the Lotus–rhizobia symbiotic relationship. The manuscript is well-organised and effectively articulates the research objectives. Consistent with prior literature, copper is shown to adversely affect root nodule symbiosis. The authors' findings suggest that these negative impacts are primarily attributable to copper toxicity influencing overall root and plant health, rather than resulting from a direct impairment of nodule function.

The evidence presented by the authors mostly supports their conclusions, however, additional experiments seem necessary to reinforce them. In addition, the ARA per nodule indicates that nodule functioning is directly impacted by Cu-treatment, revealing that this heavy metal affects the RNS both indirectly and directly.

Major comments

-Considering that the negative effects of Cu on the RNS has been widely documented, the relevance of this study is to explore at molecular level, the direct or indirect effect of this heavy metal.

-Cu treatment might interfere with the early steps of the legume-rhizobia symbiosis. For instance, what happens to root hair growth in Cu-treated plants? Lotus plants with compromised root hair growth/development are affected in IT and nodule formation. What´s the effect of Cu treatment in root hair-IT formation? IT counting is a standardized procedure to answer this question in L. japonicus.

-The ARA calculated per nodule weight indicates that nodule functioning is directly affected by Cu-treatment. In this regard, additional information could confirm or contradict this hypothesis. How is nodule occupancy by rhizobia? Or the expression of nitrogen-fixing genes?

Other comments

-The phenotype of plants should be included in the main figures, along with the plots.

-The number of biological replicates and total number of harvested plants is not clear in Material and Methods and figure legends

-Legend of Figure 1. It is not clear the number of tested/harvested plants and the number of biological replicates. It is written n=5 to n=14, depending on the panel. Five plants do not seem a robust number for solid conclusions, considering that L. japonicus is a small plant that allows high number of plants under laboratory conditions.

-

-Since Cu treatment have a negative impact on the root mas, this might explain the reduced nodule numbers as well. Plotting the nodule numbers by root mass/length would be very informative.

-Figure 3. Having representative images of nodules and nodule sections would be very informative.

Minor comments

-Please include the age plant, and days after inoculation in the figure captions.

-Readability can be improved with composite figures. For instance, Fig 2 and Fig 3 are related to nodule functioning and could be merged into a single figure.

-Line 256. “Reduced nodulation was observed in L. japonicus… “.

Ljdahps1 mutants are also defective in root hair development and nodulation.

-Line 269. “Total N2 fixation in our experiment was reduced due to the Cu-induced decrease in the number”

Should be highlighted that N2 Fixation was also significantly lower when is calculated by nodule weight. This should be discussed as well.

6. PLOS authors have the option to publish the peer review history of their article (what does this mean?). If published, this will include your full peer review and any attached files.

Reviewer #1: No

Reviewer #2: No

---

## [Author Response · Author response to Decision Letter 1]

3 Nov 2025

Response to academic editor:

We made several changes to better align with the guidelines provided by PLOS: we edited file names for figures and supporting information, changed the font of entire document to Times New Roman, centered the title on the first page, changed all equations that appear in the manuscript to be numbered and inserted via Word’s Equation Tool, and moved the Supporting Information captions to follow the references.

2. We note that there is identifying data in the Supporting Information file <S1 Table.docx>. Due to the inclusion of these potentially identifying data, we have removed this file from your file inventory.

The only identifying data we could find in the Supporting Information files were authors’ names in the metadata, so we have removed these names and changed all supplementary figures to an anonymous file format (PDF with no author). We are not sure that the anonymization process worked and would appreciate it if PLOS would check that all identifying information was removed.

Responses to Reviewer 1:

1. Line 32 - the ability to fix nitrogen should be clarified as being characteristic of bacteroids, not plants

We edited line 33 of the version with tracked changes to clarify this point. The new sentence is: “Legumes are an important source of usable nitrogen (N) in terrestrial ecosystems because of their symbiotic association with rhizobia that fix atmospheric N2 to ammonia [1].”

2. Results - the results regarding the determination of MDA and APX for shoots are missing, as these results would provide a broader, more accurate answer to the statement in the title of the paper.

We agree that shoot APX and MDA content would be beneficial in understanding the whole-plant response to Cu. However, because we initially found that Cu accumulation in response to Cu treatment was highest in the roots and nodules and less than 10% of the total Cu in the plant was found in the shoots (Fig 1), we chose to focus our experiments on the below-ground tissues. We believe that our focus primarily on the nodules is appropriately reflected in the article title: “Copper-induced stress in the legume Lotus japonicus causes loss of nitrogen fixation but not oxidative damage in nodules”.

3. Please expand abbreviations used for the first time, especially for reagents, buffers, etc., e.g., PPB.

Potassium phosphate buffer was defined in the Methods section “Leghemoglobin concentration” in line 426 in version with tracked changes, which is the first time it was used. We could not find any undefined terms.

4. Please add the equipment used for analyses.

All the equipment/materials/settings used in the experiments are now specified in the methods section with more detail. We found that several papers published by PLOS ONE also include the make/model of their equipment, so these details have also been included. The text in the relevant methods sections has been edited for clarity after these additions.

Responses to reviewer 2:

1. Considering that the negative effects of Cu on the RNS has been widely documented, the relevance of this study is to explore at molecular level, the direct or indirect effect of this heavy metal.

We thank the reviewer for appreciating the value of our study.

2. Cu treatment might interfere with the early steps of the legume-rhizobia symbiosis. For instance, what happens to root hair growth in Cu-treated plants? Lotus plants with compromised root hair growth/development are affected in IT and nodule formation. What´s the effect of Cu treatment in root hair-IT formation? IT counting is a standardized procedure to answer this question in L. japonicus.

We agree that a study of root hairs and infection threads would provide valuable additional information. We did measure the mass, number and function of nodules in our study, with the results indicating that the concentrations of Cu we used did not prevent the formation of infection threads. We are not in a position to do another study to measure root hairs and infection threads in Cu-stressed plants, but we did address the importance of these variables in lines 274-279 of the version with tracked changes: “The tips of emerging root hairs are the primary sites of entry for rhizobia and are important for the formation of infection threads [70,71]. Reduced nodulation was observed in L. japonicus mutants that produced no root hairs (rhl1 and slp) [72], mutants with altered biosynthesis pathways critical for root hair development (dahps1-1 and dahps1-2) [73], and plants with thicker and shorter root systems (rdol) [74]. Root biomass reduction, particularly fine roots and root hairs, is a characteristic response to excess Cu in many plant species [57,64].”

3. The ARA calculated per nodule weight indicates that nodule functioning is directly affected by Cu-treatment. In this regard, additional information could confirm or contradict this hypothesis. How is nodule occupancy by rhizobia? Or the expression of nitrogen-fixing genes?

We do not have data for rhizobial count/nodule occupancy. But the purpose of counting the nodules, using the ARA for nitrogenase activity and the measurements of nodule leghemoglobin were ultimately to assess nodulation and rhizobial function within those nodules. We believe this sufficiently shows the effect of Cu in these tissues. Expression of relevant genes would be very interesting and we hope future research will include this direction of inquiry, but we are not in a position to do these analyses.

4. The phenotype of plants should be included in the main figures, along with the plots.

We appreciate the reviewer’s desire to see both images and data of the plant phenotypes together, presumably in Fig 1. Representative photographs are available in the supplementary files (S2 Fig); however, we believe combining these figures would make the final image very large, becoming a 12-panel figure. If the editor would prefer, we could combine Fig 1 and S2 Fig into a single figure, or move S2 Fig into the main manuscript; we are willing to do so.

5. The number of biological replicates and total number of harvested plants is not clear in Material and Methods and figure legends. / Legend of Figure 1. It is not clear the number of tested/harvested plants and the number of biological replicates. It is written n=5 to n=14, depending on the panel. Five plants do not seem a robust number for solid conclusions, considering that L. japonicus is a small plant that allows high number of plants under laboratory conditions.

We edited the second paragraph of the Materials and Methods section (lines 388-391 in the version with tracked changes) to better explain the number of plants harvested vs. biological replicates, and what each datapoint represents. The new text is: “Because the final number of plants in each pot varied from 7 to 8, the average tissue mass, length, and nodule count of the plants in each pot was calculated. The average from each pot was then considered an independent biological replicate”.

All figure captions were also edited for clarity about the number of replicates used in each experiment. We hope the new Materials and Methods explanation makes it clearer that each replicate is not a single plant but the average of a group of plants grown in the same pot.

Some measurements, like root mass, were taken at the beginning of all experiments and so have a greater number of replicates.

6. Since Cu treatment have a negative impact on the root mass, this might explain the reduced nodule numbers as well. Plotting the nodule numbers by root mass/length would be very informative.

We agree that the reduced nodule number/mass is linked to the reduced root mass due to Cu exposure, and we addressed this in the discussion section (lines 279-281 in the version with tracked changes): “The Cu-induced reduction in root mass seen in our experiment [S3 Fig] contributed to the drop in nodulation, an observation shared by other researchers [5,63].” Because our root mass and nodule count data are already shown in Fig 1, we have not duplicated the data in a new figure in the main body of text. Instead, we have added a supplementary figure (S3 Fig) in which nodule number is plotted against root mass to further illustrate this relationship for readers.

7. Figure 3. Having representative images of nodules and nodule sections would be very informative.

We have added representative photos of nodules and nodule cross-sections from the 0 µM Cu and 400 µM Cu groups to S2 Fig. They were originally omitted due to poor picture quality.

8. Please include the age plant, and days after inoculation in the figure captions.

Figure captions have been edited to include age of plants and days after inoculation that they were harvested.

9. Readability can be improved with composite figures. For instance, Fig 2 and Fig 3 are related to nodule functioning and could be merged into a single figure.

We agree with the reviewer. We combined old Fig 2 and old Fig 3 into a single composite figure (new Fig 2). As Reviewer 2 stated, they are both ultimately measurements of nodule function and so it was appropriate to present them together. Edits were made to the caption and in-text referrals to the figure to accommodate this change. Old Fig 5 and old Fig 6 were also combined into a single composite figure (new Fig 4) as they are both measurements related to oxidative stress in the nodules and root systems. Edits were made to the caption and in-text referrals to the figure to accommodate this change.

10. Line 256. “Reduced nodulation was observed in L. japonicus…”. Ljdahps1 mutants are also defective in root hair development and nodulation.

We appreciate this information from reviewer 2, which provides another example of reduced root growth impairing nodulation. We have included a reference on dahps1 mutants in our discussion: “Reduced nodulation was observed in L. japonicus mutants that produced no root hairs (rhl1 and slp)[72], mutants with altered biosynthesis pathways critical for root hair development (dahps1-1 and dahps1-2)[73], and plants with thicker and shorter root systems (rdol) [74]” (lines 275-278 in version with tracked changes) and edited the remaining reference numbers to accommodate this change.

11. Line 269. “Total N2 fixation in our experiment was reduced due to the Cu-induced decrease in the number…” Should be highlighted that N2 Fixation was also significantly lower when is calculated by nodule weight. This should be discussed as well.

We have edited line 290 in the version with tracked changes to highlight that both number and biomass of nodules are being considered when discussing the reduction in N2 fixation. We note that this section already makes specific mention of the effect of nodule mass on N2-fixation: “Total N2 fixation in our experiment was reduced due to the Cu-induced decrease in both the number and biomass of nodules formed, and may also have been directly affected by a drop in nitrogenase activity within the nodules themselves. As nitrogenase occurs exclusively in the bacteroids within the plant’s nodules, lower nodule mass naturally results in a reduction in the total nitrogenase activity of the plant.”

---

## [Decision Letter · Decision Letter 1]

12 Dec 2025

PONE-D-25-29926R1Copper-induced stress in the legume *Lotus japonicus* causes loss of nitrogen fixation but not oxidative damage in nodulesPLOS One

Dear Dr. Macfie,

Thank you for submitting your manuscript to PLOS ONE. After careful consideration, we feel that it has merit but does not fully meet PLOS ONE’s publication criteria as it currently stands. Therefore, we invite you to submit a revised version of the manuscript that addresses the points raised during the review process.

We look forward to receiving your revised manuscript.

Kind regards,

Ying Ma, Ph.D.

Academic Editor

PLOS One

Journal Requirements:

Reviewers' comments:

Reviewer's Responses to Questions

**Comments to the Author**

1. If the authors have adequately addressed your comments raised in a previous round of review and you feel that this manuscript is now acceptable for publication, you may indicate that here to bypass the “Comments to the Author” section, enter your conflict of interest statement in the “Confidential to Editor” section, and submit your "Accept" recommendation.

Reviewer #1: All comments have been addressed

Reviewer #2: (No Response)

2. Is the manuscript technically sound, and do the data support the conclusions?

Reviewer #1: No

Reviewer #2: Yes

3. Has the statistical analysis been performed appropriately and rigorously? 

Reviewer #1: Yes

Reviewer #2: I Don't Know

4. Have the authors made all data underlying the findings in their manuscript fully available?

Reviewer #1: No

Reviewer #2: Yes

5. Is the manuscript presented in an intelligible fashion and written in standard English?

Reviewer #1: Yes

Reviewer #2: Yes

6. Review Comments to the Author

Reviewer #1: Unfortunately, the manuscript submitted for review failed to confirm the claim included in the title.

While a negative effect of copper on the nitrogen fixation process was demonstrated, only MDA and APX activity levels were determined as a reflection of oxidative stress levels – unfortunately, this is not enough to draw conclusions based on these two parameters alone.

The level of antioxidant properties of nodules and plants should be determined.

Minor comments below:

Introduction:

There is no description of the Lotus nodule type (including an anatomy and cortical tissues) and the aging process specific to this type. The information about antioxidative system in Cu-treated plants and nodules is also missing.

line 46 - "between photosynthate production" should be between the level of photosynthesis and

line 53 - remove "phytoglobin"

lines 70-71 - "Membrane disintegration, release of the bacteroid from the symbiosome" ........., This description is Unclear. Please clarify the process of symbiosome degradation in the determinate nodule, as this is the case with Lotus.

Line 79, 80 - correct the notation of units throughout the text.

Methods:

- Please expand the abbreviations used for the first time: MDA, PPB.

Results and Discussion:

The results suggest that Cu does not directly affect activity within the nodule through oxidative damage. However, Cu exposure did not demonstrate increased APX activity in the nodule, which the authors explain as "indicating a loss of antioxidant activity in the nodule in Cu-treated plants," although the results indicate that Cu is not accumulated in the nodules.

It can therefore be assumed that other components of the antioxidant system participate in reducing stress levels in the nodules, but not APX. However, the study did not demonstrate the level of the antioxidant system in the nodules what would be key in this topic. Therefore, the manuscript cannot be accepted in its current form.

Reviewer #2: (No Response)

7. PLOS authors have the option to publish the peer review history of their article (what does this mean?). If published, this will include your full peer review and any attached files.

Reviewer #1: No

Reviewer #2: No

---

## [Author Response · Author response to Decision Letter 2]

23 Jan 2026

Response to Reviewers

The line numbers mentioned in this list refer to those in the file named “Revised Manuscript with Track Changes”.

1. Unfortunately, the manuscript submitted for review failed to confirm the claim included in the title.

We have edited the title to better reflect the claims we make in our paper. We believe the new title, “Copper causes reduced nitrogen fixation but does not accumulate in the nodules of the legume Lotus japonicus” is fully supported by our data.

2. Only MDA and APX activity levels were determined as a reflection of oxidative stress levels – unfortunately, this is not enough to draw conclusions based on these two parameters alone.

We agree with the reviewer’s point that testing a broader array of antioxidants is required for definitive conclusions about the plant and nodule oxidative stress response. We address this briefly in the discussion section (lines 367-370: “No studies could be found that measured APX or any other antioxidant in the nodule in response to Cu specifically, and the narrow scope of antioxidant responses tested in this study mean that definitive conclusions about the nodule redox state cannot be drawn without further research.” and lines 378-380: “…our study did not capture a wide enough range of antioxidant activity for a fulsome picture of antioxidant responses to Cu-induced ROS in the roots and nodules”.

We have also made a minor edit to the results section (lines 271-272 were deleted remove the suggestion that our data can be used to indicate antioxidant activity.

3.

a. There is no description of the Lotus nodule type (including an anatomy and cortical tissues) and the aging process specific to this type.

We have added some information about determinate and indeterminate nodule morphologies (lines 44-48: “Nodules primarily fall under two morphologies: the indeterminate and determinate nodule [12, 13]. Indeterminate nodules are characterized by persistent meristematic activity and create cylindrical nodules with specific zones of activity. Determinate nodules (such as those seen in L. japonicus) lose meristematic activity after rhizobial infection and an initial period of cell division, forming spherical nodules with fairly homogenous inner tissue [9,12,14].)

We also added information about their different aging/senescence processes (lines 49-63: “Nodule senescence is a regulated process that can be considered the final step in the lifespan of the nodule [9,15,16]. N2 fixation activity peaks at approximately 4 weeks, after which activity steadily declines [15,16]. Symbiosome membrane disintegration begins and lytic symbiosome compartments are formed in order to facilitate nutrient remobilization and reabsorption processes, ultimately ending in the death of the host cell [9,17]. In unstressed nodules, cells senesce sequentially from oldest to youngest, meaning that in indeterminate nodules it progresses from zone IV outwards, and progresses radially from the center in determinate nodules [11,16]. In determinate nodules, bacteroids do not terminally differentiate and are released during senescence, able to revert to a free-living lifestyle [9,11]. Stressors like defoliation, dark, and drought can induce rapid early senescence, as can available N in soil [9,17,18]. Senescence is generally characterized by activation of proteolytic enzymes, a decrease in cytosolic proteins [11,17], and by shifts to the nodule redox balance due to the formation of reactive oxygen species (ROS) and changing activity of nodule antioxidants [17,19]. Some nodule antioxidant enzymes are reported to upregulate during senescence, while others decrease or do not vary significantly [15].”).

We do not address nodule internal anatomy in detail, as we did not separate the internal tissues of the nodules for any experiments, and cannot make any conclusions specific to a nodule tissue type.

b. The information about antioxidative system in Cu-treated plants and nodules is also missing.

We have added additional information on nodule antioxidants to two places in the introduction section. First, to the paragraph concerning plant and nodule antioxidant systems (lines 83-89: “Nodule antioxidants are similar to those found in the rest of the plant but are present at higher concentrations to compensate for O2 produced during the highly energetic process of N2 fixation [30,31]. The most abundant non-enzymatic antioxidants of the nodule are ascorbate and glutathione, whose concentrations in the nodule are positively correlated with N2 fixation [30]. A non-exhaustive list of enzymatic nodule antioxidants includes SOD, catalase (CAT), thiol peroxidases, and the enzymes of the ascorbate-glutathione pathway [26,32-34].”).

Second, to the paragraph about copper toxicity in plants (lines 128-137: “Alfalfa seedlings exposed to 0 to 400 mg/kg Cu had reduced biomass, tissue N concentrations, and nodule number and weight at higher Cu concentrations. The antioxidant response to Cu-induced ROS is more variable; activity of SOD in the roots decreased as Cu increased, while activity of APX and CAT increased with Cu until declining at higher Cu treatments [58]. Another experiment on alfalfa showed ROS production in the roots increased when they were exposed to Cu, triggering an increase in the activities of APX, CAT and glutathione reductase at lower Cu concentrations but a decrease in antioxidant activity at higher concentrations [59]. Soybean grown in 0 to 100 µM Cu showed reduced nodule number and size, and a similar dose-dependent response in antioxidants in response to Cu [60].”).

We hope this elaborates sufficiently on antioxidant systems in the nodule throughout its lifecycle and clarifies the specific impact of copper on nodule function and nodular antioxidant systems.

4. line 46 - "between photosynthate production" should be “between the level of photosynthesis and”

We have made edits to the lines 67-69, which now read: “…new nodule formation must be regulated to maintain equilibrium between energy production via photosynthesis in the shoots and energy consumption in the nodules [14].”

5. line 53 - remove "phytoglobin"

The suggested edit has been made (line 75).

6. lines 70-71 - "Membrane disintegration, release of the bacteroid from the symbiosome" ........., This description is Unclear. Please clarify the process of symbiosome degradation in the determinate nodule, as this is the case with Lotus.

This section (lines 49-63) has been expanded to better explain the stages of senescence, and the unique characteristics of determinate nodules in particular (see the second paragraph of our response to point 3a).

7. Line 79, 80 - correct the notation of units throughout the text.

Notation of units have been edited throughout the text (e.g., from mg kg-1 to mg/kg) to conform to unit styles we observed most frequently in other PLOS One papers.

8. Please expand the abbreviations used for the first time: MDA, PPB.

All abbreviations are defined in the text at their first use.

9.

a. The results suggest that Cu does not directly affect activity within the nodule through oxidative damage. However, Cu exposure did not demonstrate increased APX activity in the nodule, which the authors explain as "indicating a loss of antioxidant activity in the nodule in Cu-treated plants," although the results indicate that Cu is not accumulated in the nodules.

We believe that the last sentence of our results section may have misled the reviewer. As the reviewer states, our results indicate that Cu does not accumulate in the nodules and does not directly affect nodule activity. We found no evidence of lipid peroxidation in nodules with increased Cu treatment, further indicating that Cu is not causing oxidative damage directly to the nodules. We have edited the last sentence of the results section to focus solely on the numerical response to experimental treatment. We have also elaborated on our explanation of Cu-induced changes in APX activity in the discussion section (lines 357-380). We suggest that reduced APX activity in nodules is due to Cu stress in the roots and shoots of the plant, in turn inducing premature senescence of the nodules.

b. It can therefore be assumed that other components of the antioxidant system participate in reducing stress levels in the nodules, but not APX. However, the study did not demonstrate the level of the antioxidant system in the nodules what would be key in this topic. Therefore, the manuscript cannot be accepted in its current form.

We agree with the reviewer that a complete antioxidant profile would allow us to draw more concrete conclusions about the response to Cu-induced ROS in the entire plant, including nodules, and have addressed this criticism in point 2 above. It is very likely that antioxidants that were not measured in our study are active. However, we believe the assumption that other nodule antioxidants must be the key to the nodule’s response is incorrect. The lack of Cu accumulation or measurable oxidative damage in the nodules themselves strongly supports our primary finding that Cu’s effect on N fixation is not due to a direct effect of Cu-induced oxidation in the nodules.

Additional changes:

We have made some changes to the order of paragraphs in the introduction, particularly the paragraph about nodule senescence. Along with the edits to the paragraph contents, we believe this change increases the readability and overall flow of the introduction section.

We made appropriate edits to the in-text citations and references section in accordance with edits made to the text.

We have removed our list of abbreviations to conform to the style we observed most frequently in other PLOS One papers.

---

## [Decision Letter · Decision Letter 2]

17 Mar 2026

PONE-D-25-29926R2Copper causes reduced nitrogen fixation but does not accumulate in the nodules of the legume *Lotus japonicus*PLOS One

Dear Dr. Macfie,

Thank you for submitting your manuscript to PLOS ONE. After careful consideration, we feel that it has merit but does not fully meet PLOS ONE’s publication criteria as it currently stands. Therefore, we invite you to submit a revised version of the manuscript that addresses the points raised during the review process.

We look forward to receiving your revised manuscript.

Kind regards,

Ying Ma, Ph.D.

Academic Editor

PLOS One

**Journal Requirements:**

Reviewers' comments:

Reviewer's Responses to Questions

**Comments to the Author**

1. If the authors have adequately addressed your comments raised in a previous round of review and you feel that this manuscript is now acceptable for publication, you may indicate that here to bypass the “Comments to the Author” section, enter your conflict of interest statement in the “Confidential to Editor” section, and submit your "Accept" recommendation.

Reviewer #3: All comments have been addressed

2. Is the manuscript technically sound, and do the data support the conclusions?

Reviewer #3: Yes

3. Has the statistical analysis been performed appropriately and rigorously? 

Reviewer #3: Yes

4. Have the authors made all data underlying the findings in their manuscript fully available?

Reviewer #3: Yes

5. Is the manuscript presented in an intelligible fashion and written in standard English?

Reviewer #3: Yes

6. Review Comments to the Author

Reviewer #3: In the present manuscript, the authors evaluated the effect of increasing Cu addition on plant growth, nodule activity, and N2 fixation in L. japonicus. The manuscript is very well written. The Materials and Methods section describes in detail the methodologies used in the experiment, as well as the measurement protocols, which is highly appreciated. The statistical analysis is well performed and appropriate for evaluating the effect of the treatments applied.

One point of critique is that the manuscript places strong emphasis on oxidative processes occurring within the nodules and on the associated measurements. In my view, the interpretation of the experiment could be simpler. The addition of Cu affected plant growth, likely reducing photosynthesis, which in turn reduced plant N demand and consequently decreased the supply of assimilates to the nodules. This reflects the well-known plant–microsymbiont trade-off (cost–benefit relationship). Importantly, this phenomenon is not only observed under toxicity conditions but also under P deficiency. Deficiency of other nutrients can also induce nodule senescence, increased leghemoglobin degradation (leading to the green coloration of nodules), and reduced nodule functionality. A decrease in O2 transport to symbiosomes is likely a primary mechanism inhibiting N2 fixation when plant N demand decreases. The observations within the nodules (e.g., reduced leghemoglobin levels) may therefore be consequences of reduced N demand rather than direct oxidative damage. Therefore, this interpretation does not contradict the presented data. I agree with author´s conclusions.

I believe the manuscript would benefit from shortening the arguments related to oxidative processes, particularly since Cu concentration did not increase in nodules. This may represent the main unresolved point of the study: explaining why Cu concentration did not increase in nodules (for example, via qPCR analysis of metal transporters). However, this aspect is partially addressed in the discussion.

Overall, with appropriate refinements to the Introduction, Results, and Discussion sections, the manuscript will present a fully coherent narrative. I agree with the revisions made during previous review rounds and with the authors’ responses to reviewers.

Minor comments related to the Results section:

Consider moving some text currently in the Results to the Discussion (lines 133–135, 188–189, 212–218).

Best regards

7. PLOS authors have the option to publish the peer review history of their article (what does this mean?). If published, this will include your full peer review and any attached files.

Reviewer #3: No

---

## [Author Response · Author response to Decision Letter 3]

20 Apr 2026

Response to Reviewers

Journal Requirements:

Not applicable

We did not find any retracted papers in our reference list. However, we removed the old reference number 1 because we could not verify its doi . We also removed old reference numbers 9, 17,18, 22, 31, 37, 38 and 53 because they were redundant. We added one new reference (new # 8). The reference list has been checked for completeness, and we have corrected some minor errors.

Reviewer #3

1. In the present manuscript, the authors evaluated the effect of increasing Cu addition on plant growth, nodule activity, and N2 fixation in L. japonicus. The manuscript is very well written. The Materials and Methods section describes in detail the methodologies used in the experiment, as well as the measurement protocols, which is highly appreciated. The statistical analysis is well performed and appropriate for evaluating the effect of the treatments applied.

Thank you for these positive comments.

2. One point of critique is that the manuscript places strong emphasis on oxidative processes occurring within the nodules and on the associated measurements. In my view, the interpretation of the experiment could be simpler. The addition of Cu affected plant growth, likely reducing photosynthesis, which in turn reduced plant N demand and consequently decreased the supply of assimilates to the nodules. This reflects the well-known plant–microsymbiont trade-off (cost–benefit relationship). Importantly, this phenomenon is not only observed under toxicity conditions but also under P deficiency. Deficiency of other nutrients can also induce nodule senescence, increased leghemoglobin degradation (leading to the green coloration of nodules), and reduced nodule functionality. A decrease in O2 transport to symbiosomes is likely a primary mechanism inhibiting N2 fixation when plant N demand decreases. The observations within the nodules (e.g., reduced leghemoglobin levels) may therefore be consequences of reduced N demand rather than direct oxidative damage. Therefore, this interpretation does not contradict the presented data. I agree with author´s conclusions.

I believe the manuscript would benefit from shortening the arguments related to oxidative processes, particularly since Cu concentration did not increase in nodules. This may represent the main unresolved point of the study: explaining why Cu concentration did not increase in nodules (for example, via qPCR analysis of metal transporters). However, this aspect is partially addressed in the discussion.

Overall, with appropriate refinements to the Introduction, Results, and Discussion sections, the manuscript will present a fully coherent narrative. I agree with the revisions made during previous review rounds and with the authors’ responses to reviewers.

We thank the reviewer for these suggestions. In our response below, we have used line numbers from the version with tracked changes.

Introduction:

a. Lines 50-53. We included factors other than oxidative stress that can induce senescence. It now reads: “Rapid early senescence can be induced when the production of sugars by the host plant is limited due to stressors such as defoliation, phosphorus deficiency, darkness and drought [reviewed in 15], or when available N in soil increases [14].

b. We deleted lines 57-58, in which we went into more detail than required about antioxidant enzymes: “ Some nodule antioxidant enzymes are reported to upregulate during senescence, while others decrease or do not vary significantly”.

c. Lines 73-74. We moved this line from the results section “The presence of Lb in the nodules is closely associated with nodule function because of its role shuttling O2 to the bacteroids”. We placed it in the introduction to build the case for using leghemogloblin concentration as a measure of nodule function.

d. Line 78. We added a line “Leghemoglobin degrades during nodule senescence” to provide more background for our conclusion that our results indicate nodule senescence due to plant stress.

e. lines 81-84. We deleted these because they provided more detail than necessary about antioxidants: “The most abundant non-enzymatic antioxidants of the nodule are ascorbate and glutathione, whose concentrations in the nodule are positively correlated with N2 fixation [30]. A non-exhaustive list of enzymatic nodule antioxidants includes superoxide dismutase (SOD), catalase (CAT), thiol peroxidases, and the enzymes of the ascorbate-glutathione pathway [26,32,34].

f. Lines 114-119. We deleted information about changes in antioxidant enzyme activity in response to copper: “…activity of SOD in the roots decreased as Cu increased, while activity of APX and CAT increased with Cu until declining at higher Cu treatments [58]. Another experiment on alfalfa showed ROS production in the roots increased when they were exposed to Cu, triggering an increase in the activities of APX, CAT and glutathione reductase at lower Cu concentrations but a decrease in antioxidant activity at higher concentrations [59].” We kept a few words about APX activity to build our case for measuring it in our study.

Results

g. Lines 142-143: We deleted this line because the idea was already in the discussion. Discussion: “Root biomass reduction, particularly fine roots and root hairs, is a characteristic response to excess Cu in many plant species [57, 66].”

h. Lines 196-197. As mentioned in point 3 above, we moved this line to the introduction (lines 73-74): “The presence of Lb in the nodules is closely associated with nodule function because of its role shuttling O2 to the bacteroids”.

i. Lines 220-223: We integrated “This would indicate that while copper was taken up by the plants, it did not accumulate in the nodules in proportion to its availability despite the nodules being immediately adjacent to the roots and having a connected vasculature [68]. This would indicate that Cu transport to the nodule tissues is restricted in some fashion so that internal Cu concentration remains relatively constant” into lines 304-308 of the discussion: “Despite the evidence that excess Cu inhibited nodule development and activity, measurements of tissue Cu concentrations revealed that Cu did not accumulate in the nodule tissues, even though the nodules are immediately adjacent to the roots and have a connected vasculature [72]. This would indicate that Cu transport to the nodule tissues is restricted in some fashion so that internal Cu concentration remains relatively constant”.

j. Lines 223-226. We deleted “It is also therefore unlikely that the differences in nodulation and N2 fixation observed are due to Cu acting directly on the nodule interior” and added “Differences in nodulation and N2 fixation observed were not due to accumulation of Cu in the nodule interior” to lines 365-366 of the concluding paragraph.

Discussion

See points i and j above.

3. Consider moving some text currently in the Results to the Discussion (lines 133–135, 188–189, 212–218).

Done, see points g toj above.

Response to the editor

We checked our figures against the journal’s guidelines and submitted them to the NAAS site for evaluation. The NAAS software re-sized the figures to be smaller than the journal’s listed figure dimensions by cropping parts of the figures. So, we re-sized the figures to match the output from NAAS. When we re-submitted these smaller figures to NASS, it further reduced the figure dimensions and cropped the margins around each figure, bringing the text of the X and Y axes to the very edge of the field and making them difficult to read. We have opted to submit figures that have slightly smaller dimensions than required by the journal but, to keep the white space (margins) around each figure, do not pass the NAAS assessment. If you require even smaller figures, we would be happy to provide them.

---

## [Decision Letter · Decision Letter 3]

26 Apr 2026

Copper causes reduced nitrogen fixation but does not accumulate in the nodules of the legume *Lotus japonicus*

PONE-D-25-29926R3

Dear Dr. Macfie,

We’re pleased to inform you that your manuscript has been judged scientifically suitable for publication and will be formally accepted for publication once it meets all outstanding technical requirements.

Kind regards,

Ying Ma, Ph.D.

Academic Editor

PLOS One

Additional Editor Comments (optional):

Reviewers' comments:

Reviewer's Responses to Questions

**Comments to the Author**

1. If the authors have adequately addressed your comments raised in a previous round of review and you feel that this manuscript is now acceptable for publication, you may indicate that here to bypass the “Comments to the Author” section, enter your conflict of interest statement in the “Confidential to Editor” section, and submit your "Accept" recommendation.

Reviewer #3: All comments have been addressed

2. Is the manuscript technically sound, and do the data support the conclusions?

Reviewer #3: Yes

3. Has the statistical analysis been performed appropriately and rigorously? 

Reviewer #3: Yes

4. Have the authors made all data underlying the findings in their manuscript fully available?

Reviewer #3: Yes

5. Is the manuscript presented in an intelligible fashion and written in standard English?

Reviewer #3: Yes

6. Review Comments to the Author

Reviewer #3: Dear Editor and authors,

In the revised version of the manuscript, the authors have adequately addressed the comments and suggestions raised by this reviewer. The article now presents a coherent structure, with a clear line of reasoning that is well supported by the results obtained.

In my opinion, the manuscript has significantly improved and is now suitable for publication.

Sincerely

7. PLOS authors have the option to publish the peer review history of their article (what does this mean?). If published, this will include your full peer review and any attached files.

Reviewer #3: **Yes:** Ricardo Cabeza

---

## [Editor Report · Acceptance letter]

PONE-D-25-29926R3

PLOS One

Dear Dr. Macfie,

I'm pleased to inform you that your manuscript has been deemed suitable for publication in PLOS One. Congratulations! Your manuscript is now being handed over to our production team.

Kind regards,

on behalf of

Dr. Ying Ma

Academic Editor

PLOS One